# The Mediating Role of Loneliness and the Moderating Role of Gender between Peer Phubbing and Adolescent Mobile Social Media Addiction

**DOI:** 10.3390/ijerph191610176

**Published:** 2022-08-17

**Authors:** Xiao-Pan Xu, Qing-Qi Liu, Zhen-Hua Li, Wen-Xian Yang

**Affiliations:** 1Institute for Public Policy and Social Management Innovation, College of Political Science and Public Administration, Henan Normal University, Xinxiang 453007, China; 2School of Sociology, Central China Normal University, Wuhan 430079, China; 3College of Education for the Future, Beijing Normal University, Zhuhai 519087, China; 4School of Education, Guangzhou University, Guangzhou 510006, China; 5Center of Mental Health Education and Counseling, Lanzhou University, Lanzhou 730000, China

**Keywords:** peer phubbing, mobile social media addiction, loneliness, gender, adolescents

## Abstract

Mobile social media addiction has been a pressing issue in adolescents. The present study examined the mediation of loneliness between peer phubbing and mobile social media addiction among Chinese adolescents and tested whether gender could moderate the direct and indirect effects of peer phubbing. A total of 830 adolescents between 11 and 18 years of age (*M*_age_ = 14.480, *SD*_age_ = 1.789) completed an anonymous self-report survey. The results showed that peer phubbing was positively associated with mobile social media addiction. Loneliness partially mediated peer phubbing and adolescent mobile social media addiction. There were significant gender differences in the direct and indirect effects of peer phubbing on mobile social media addiction. The direct effect of peer phubbing and the indirect effect through loneliness were relatively higher in girls than in boys. The results highlight the critical role of loneliness in linking peer phubbing to adolescent mobile social media addiction and the vital role of gender in moderating the direct and indirect impacts of peer phubbing. The findings promote a better understanding of how peer phubbing is associated with adolescent mobile phone addiction and for whom the effect of peer phubbing is potent.

## 1. Introduction

Social media addiction has been conceptualized as a typical behavioral addiction [1,2,3]. Mobile social media addiction, a sub-type of mobile phone addiction, refers to a state in which an individual is excessively addicted to mobile social network services, resulting in a negative impact on physical and mental functions [4,5]. Different from other terms, such as problematic social media use or compulsive social media use, mobile social media addiction was used to highlight that these behaviors of overusing social network services on mobile phones have four typical symptoms of mobile phone addiction (i.e., loss of control, anxiety and feeling lost, withdrawal and escape, and productivity loss) [4,5,6,7]. Adolescents are a high incidence group for mobile phone addiction [8]. Since adolescents generally have strong belonging needs in peer relationships [9], they are also particularly prone to develop mobile social media addiction. Some studies have confirmed the negative effects of mobile social media addiction on individual development. For instance, Błachnio, Przepiorka, and Pantic found that Facebook addiction was positively associated with low self-esteem and life satisfaction [10]. Xue et al. revealed that addiction to WeChat (a mobile app with dominant usage in China) was negatively correlated with users’ physical, mental, and social health [5]. Therefore, it is of great significance to explore the influencing factors and inner mechanisms of adolescent mobile social media addiction. The findings would help deepen the understanding of adolescents’ mobile social media addiction and provide practical implications for the prevention and intervention of mobile social media addiction in adolescents.

Peer relationships have a critical impact on adolescent mobile phone addiction [9]. Peer phubbing refers to the state that individuals feel ignored by peers because their peers pay more attention to mobile phones [11]. Peer phubbing is a typical manifestation of phubbing in peer interactions. Phubbing widely appears in people of different ages and has a significantly negative impact on individuals’ interpersonal relationships and mental health. Previous research has illustrated that partner phubbing had a negative influence on romantic relationship satisfaction [12,13]. Parental phubbing had significant direct and indirect effects on adolescent mobile phone addiction [14,15]. Peer phubbing could increase the risk of mobile phone addiction among college students [11,16]. However, few studies have explored the effect of peer phubbing on adolescent mobile phone addiction and the underlying mechanisms. Testing the effect of peer phubbing on adolescent mobile social media and its mechanisms could provide scientific suggestions for alleviating the impact of peer phubbing and reducing the risk of mobile social media addiction in adolescents.

Loneliness is a negative experience in which interpersonal relationships cannot meet individuals’ needs in quality and quantity [17]. According to the Compensatory Internet Use theory [18], people who cannot meet their psychological needs in the real world seek compensation through the Internet. Individuals with high levels of loneliness, due to unsatisfied needs tend to alleviate negative emotions by overusing the Internet and mobile phones [19,20]. Many previous studies have confirmed that loneliness can strengthen the risk of adolescent mobile phone addiction [21,22,23]. The effect of loneliness on social media addiction has also been demonstrated by prior research [24,25]. Therefore, adolescents who experience high levels of loneliness may engage more in mobile social networking services to meet their psychological needs. 

In addition, peer phubbing may be an essential predictor of adolescent loneliness. Studies have confirmed that peer rejection was one of the direct factors of adolescent loneliness. For instance, Xiao, Bullock, Liu, and Coplan found that peer rejection significantly contributed to the increase in loneliness [26]. Clear, Zimmer-Gembeck, Duffy, and Barber revealed that more frequent experience of peer victimization and rejection was associated with more loneliness [27]. Peer phubbing makes adolescents have a negative experience of being rejected and ignored, which will undoubtedly aggravate their feelings of loneliness. Since peer phubbing may enhance loneliness and loneliness, in turn, will aggravate adolescents’ mobile phone addiction, we hypothesized that loneliness would play a mediating role in the association between peer phubbing and adolescent mobile phone addiction.

Moreover, gender may have an apparent impact on the mediation model of loneliness. There are significant gender differences in loneliness and mobile phone addiction. Previous studies have shown that females tend to experience more loneliness than males [28,29] and are more likely to become addicted to mobile phones because of interpersonal relationships [30,31]. Therefore, girls who experience peer phubbing may have high levels of loneliness and risk of mobile social media addiction. We thus hypothesized that gender may moderate the direct effect of peer phubbing and its indirect effect through loneliness. Specifically, compared with boys, the direct effect of peer phubbing on adolescents’ mobile social media addiction and the mediating effect of loneliness may be stronger in girls.

In conclusion, the present study focused on the effect of peer phubbing on adolescents’ mobile social media addiction and the underlying mechanisms. We aimed to address two main hypotheses: (a) would loneliness act as a mediator linking peer phubbing to mobile social media addiction in general for all groups and in a disaggregated sense, and (b) would gender act as a moderator regulating the direct and indirect effects of peer phubbing. The mediation hypothesis examined the path through which peer phubbing was correlated with adolescent mobile social media addiction. The moderating hypothesis examined the condition when, or for whom, the relationship between peer phubbing and adolescent mobile social media addiction was potent. The results would help uncover how peer phubbing is associated with adolescent mobile phone addiction, and whether this applies to all people, or only to certain gender groups, in which the direct and indirect effects of peer phubbing may be potent.

## 2. Materials and Methods

### 2.1. Participants and Procedure

The present study was approved by the Institutional Review Board at the first author’s affiliation. Informed consent was obtained from the schools, individual participants, and guardians. A total of 896 adolescents from two high schools in South China were invited to participate in our anonymous survey in the classrooms during normal school lessons. After excluding 66 adolescents who failed to complete all the questionnaires or provided invalid responses, data of 830 adolescents were included in our formal analysis. Among these participants, 453 adolescents (54.60%) were girls, and 377 were boys (45.40%). The mean age of the participants was 14.480 (*SD*_age_ = 1.789).

### 2.2. Measurements

#### 2.2.1. Peer Phubbing

Peer phubbing was measured by a nine-item scale adapted from the Partner Phubbing Scale [13]. Participants answered these nine items (e.g., “My peers use mobile phones when we are out together”) on a five-point scale (1 = never, 5 = always). High scores indicated high levels of peer phubbing. Cronbach’s α for this measure was 0.858.

#### 2.2.2. Loneliness

The Chinese version [32] of the UCLA Loneliness Scale [33] was used. It consists of 20 items (e.g., “How often do you feel that you are no longer close to anyone”) rated on a four-point scale (1 = never, 4 = always). High scores indicated high levels of loneliness. Cronbach’s α for this measure was 0.939.

#### 2.2.3. Mobile Social Media Addiction

Mobile social media addiction was measured by the Mobile Social Addiction subscale of the Mobile Phone Addiction Type Scale (MPATS) developed for Chinese adolescents and young adults [4]. It consists of six items rated on a five-point scale. Sample items are “Every chance I get, I open the social networking apps on my phone, even if it is just for a few glances” and “I overlook interacting with my family or friends because I spend too much time socializing on my phone”. High scores indicated high levels of mobile social media addiction. Cronbach’s α for this measure was 0.924.

### 2.3. Main Statistical Analyses

The descriptive statistics, the independent sample *t*-test, and the Pearson correlation analysis were conducted to reflect the intercorrelations between the core variables using the statistical software package SPSS 23.0 (IBM Corporation, Armonk NY, USA). The PROCESS macro for SPSS was used to conduct the mediation model and the moderated mediation model analysis [34]. The mediation model analysis was performed to test the mediating role of loneliness. The moderated mediation model analysis was performed to test the moderating role of gender on the direct and indirect effects of peer phubbing. Age and daily mobile phone use were included as covariates because of their potential impacts on mobile social media addiction [35,36].

## 3. Results

### 3.1. Preliminary Analysis

Significant differences were found in the scores of loneliness and mobile social media addiction (see Table 1). Specifically, girls scored higher on loneliness (*t* = 4.41, *p* < 0.001) and mobile social media addiction (*t* = 3.415, *p* < 0.01) than boys. The results of the Pearson correlation analysis are presented in Table 2. In both boy and girl groups, peer phubbing, loneliness, and mobile social media addiction were significantly correlated.

### 3.2. Testing for the Mediation Model of Loneliness

We conducted a mediation model analysis to test the role of loneliness in the association between peer phubbing and adolescent mobile social media addiction. Table 3 presents the results generated by the Model 4 of the PROCESS macro for SPSS [34]. After controlling for age and daily mobile phone use time, the direct path coefficient from peer phubbing to mobile social media addiction in the absence of the mediator (i.e., loneliness) was significant (β = 0.453, *p* < 0.001). In addition, peer phubbing significantly predicted loneliness (β = 0.339, *p* < 0.001). When peer phubbing and loneliness were included as the predictors together, loneliness significantly predicted adolescent mobile social media addiction (β = 0.252, *p* < 0.001), and the effect of peer phubbing on mobile social media addiction was still significant (β = 0.225, *p* < 0.001). The bias-corrected percentile bootstrap method showed that the mediating effect of loneliness was 0.116, and its 95% confidence interval was 0.080, 0.159. The mediation effect of social anxiety accounted for 33.69% of the total effect. In other words, loneliness played a weak but significant mediating effect in the association between peer phubbing and mobile social media addiction.

### 3.3. Testing for the Moderated Mediation Model of Loneliness and Gender

We conducted a moderated mediation model analysis to test whether gender could moderate the mediation of loneliness in the association between peer phubbing and mobile social media addiction. Table 4 presents the results generated by the Model 8 of the PROCESS macro for SPSS. After controlling for age and daily mobile phone use time, peer phubbing significantly predicted loneliness (β = 0.448, *p* < 0.001), and this effect was moderated by gender (β = 0.300, *p* < 0.001). The association between peer phubbing and loneliness was stronger for girls than for boys. The plot of the relationship between peer phubbing and loneliness in boys and girls is depicted in Figure 1. In addition, peer phubbing significantly predicted mobile social media addiction (β = 0.238, *p* < 0.001), and this effect was moderated by gender (β = 0.296, *p* < 0.001). The association between peer phubbing and mobile social media addiction was moderate in girls (β = 0.373, *p* < 0.001), but almost meaningless in boys (β = 0.007, *p* > 0.05). The plot of the relationship between peer phubbing and mobile social media addiction in boys and girls is depicted in Figure 2.

Moreover, the conditional indirect effect analysis showed that the mediating effect of loneliness was moderated by gender. The mediating effect of loneliness was relatively higher in girls (β = 0.124, *p* < 0.01) than in boys (β = 0.06, *p* < 0.01).

## 4. Discussion

Although the influencing factors of general mobile phone addiction have been explored in many studies, there are still relatively few studies focusing on the specific types of mobile phone addiction [4]. The present study explored the influencing factors and mechanisms of mobile social media addiction in adolescents. The results showed that loneliness played a mediating role in the association between peer phubbing and adolescent mobile social media addiction. In addition, gender could moderate the direct and indirect effects of peer phubbing. The direct effect of peer phubbing and the indirect effect through loneliness were stronger in girls than in boys. The present study extends previous findings on mobile phone addiction and contributes to the prevention and intervention of mobile social media addiction in adolescents.

This study verified that peer phubbing was positively correlated with mobile social media addiction among adolescents. This result was in accordance with previous research showing that peer phubbing was strongly associated with mobile phone addiction among college students [11]. The innovation of our findings is that we tested the effect of peer phubbing on the specific type of mobile phone addiction (i.e., mobile social media addiction) among adolescents. Furthermore, we also uncovered the psychological mechanisms underlying the association between peer phubbing and adolescent mobile social media addiction.

We found that loneliness partially mediated the association between peer phubbing and adolescent mobile social media addiction. In other words, peer phubbing was positively correlated with adolescent loneliness, which was, in turn, positively correlated with mobile social media addiction. Although previous research has demonstrated that peer rejection was closely associated with feelings of loneliness [26,27], no previous studies have tested the effect of peer phubbing. In the mobile internet era, peer phubbing has been the typical behavior of peer rejection. Our result expands previous findings by indicating that, in addition to direct peer rejection and exclusion, indirect rejection and exclusion behaviors, such as peer phubbing, also strengthen the loneliness experience. For the second stage of the mediation model (i.e., the association between loneliness and mobile social media addiction), our result was consistent with previous studies verifying the impact of loneliness on mobile phone addiction [21,22,23]. However, our result revealed in detail that the increased risk caused by loneliness is mobile social media addiction. Overall, the mediation model of loneliness in the association between peer phubbing and mobile social media addiction also coincides with the Compensatory Internet Use theory [18]. Adolescents who are rejected due to peers’ overuse of mobile phones may have more unmet relatedness needs and stronger loneliness experiences, which encourages them to seek alternative compensations with the help of the Internet. In the era of mobile Internet, various social applications on mobile phones provide individuals with great convenience. Many adolescents may use mobile social applications to establish and maintain interpersonal relationships, leading to a significant addiction to mobile social applications.

The present study also found significant gender differences in the direct and indirect effects of peer phubbing. The direct effect of peer phubbing on mobile social media addiction was relatively higher in girls than in boys (the effect was not significant in boys). The indirect effect of peer phubbing on mobile social media addiction through loneliness was also relatively higher in girls than in boys (the indirect effects in girls and boys were both significant). In other words, high levels of peer phubbing were associated with mobile social media addiction directly and indirectly through loneliness in girls, but not in boys. Previous studies have found that females had a higher level of mobile phone addiction than males [30,37], and females preferred online social interaction over males [38,39]. In line with these results, girls in our study had relatively higher levels of mobile social media addiction than males. Since females have stronger relatedness needs [39,40] and higher rejection sensitivity [41,42], when facing interpersonal relationship problems (such as peer phubbing), they may avoid problems and alleviate negative emotions more through mobile social interaction. Females are also more likely to engage in indirect forms of bullying (e.g., ignoring, rejecting, or relational aggression) victimization [43,44]. Peer phubbing may be a novel manifestation of the indirect form of bullying victimization. Furthermore, interpersonal relationship problems may cause females to develop a stronger sense of loneliness, indirectly encouraging them to engage more in mobile social networking services.

The present study has some limitations. First, the cross-sectional design we used cannot strictly confirm the causality of the mediation model. Future research may consider adopting a longitudinal design or experimental study. Second, the present study only focused on one type of mobile phone addiction and did not compare the potential effects of peer phubbing on different types of mobile phone addiction. Recent research has developed a Mobile Phone Addiction Type Scale to distinguish mobile phone addiction into four types: mobile social media addiction, mobile game addiction, mobile information acquisition addiction, and mobile short-form video addiction [1]. Future research may examine whether peer phubbing has different effects on different types of mobile phone addiction. Third, there might be several paths through which peer phubbing is linked to mobile social media addiction. In our study, loneliness only partially mediated the effect of peer phubbing on mobile social media addiction. Future research may test other potential mediators closely correlated with adolescent mobile social media addiction. Fourth, given the weak categorical associations (of males and females) in our study, other factors on the macro-level of the school environment or the micro-level of individual psychology may be a better route to helping people deal with peer phubbing and mobile social media addiction. Future research may explore other protective factors with strong buffering effects to generate more practical interventions. 

Despite some limitations, the present study still has important theoretical and practical implications. From the theoretical perspective, this study not only verifies the effect of peer phubbing on mobile social media addiction, but also uncovers the mechanisms underlying the association between peer phubbing and adolescent mobile social media addiction. The findings can answer how peer phubbing is associated with adolescent mobile social media addiction and for whom the direct and indirect effects of peer phubbing are potent. From a practical perspective, this study can provide specific suggestions for the prevention and intervention of mobile social media addiction among adolescents. Families, schools, and communities should play a guiding role in organizing group activities for adolescents. For instance, teachers may promote close communication among peers and reduce peer phubbing through class meetings on multiple themes. Communities can help teenagers establish and develop peer relationships by building a child-friendly community and carrying out peer group activities, such as team sports or puzzle games. Teachers and parents should also reduce adolescents’ feelings of loneliness and weaken the risk of mobile phone addiction through regular emotional communication (such as heart-to-heart talks between teachers and students) and various interesting activities (such as short-distance family tourism). In addition, since peer phubbing and loneliness resulting from peer phubbing have a more potent impact on girls, teachers and parents should pay special attention to girls’ peer relationships. They can tell girls some peer communication strategies and conflict resolution methods so that girls can better establish and maintain peer relationships. 

## 5. Conclusions

Peer phubbing was positively associated with adolescent mobile social media addiction. Loneliness acted as a mediator. The direct effect of peer phubbing and the mediating effect of loneliness were moderated by gender, in that these two effects were relatively more potent in girls than in boys (the direct effect was almost nonexistent in boys). The findings highlight how peer phubbing is directly and indirectly associated with adolescent mobile social media addiction in general, and how the direct and indirect effects of peer phubbing are more potent in girls than in boys, when comparing averages of both gender groups.

## Figures and Tables

**Figure 1 ijerph-19-10176-f001:**
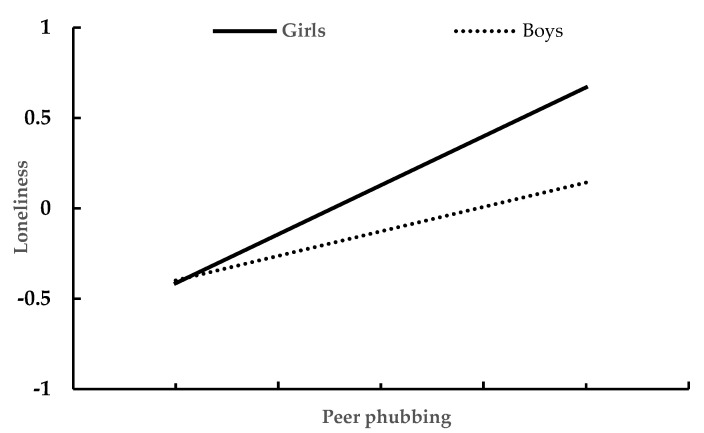
The relationship between peer phubbing and loneliness in girls and boys.

**Figure 2 ijerph-19-10176-f002:**
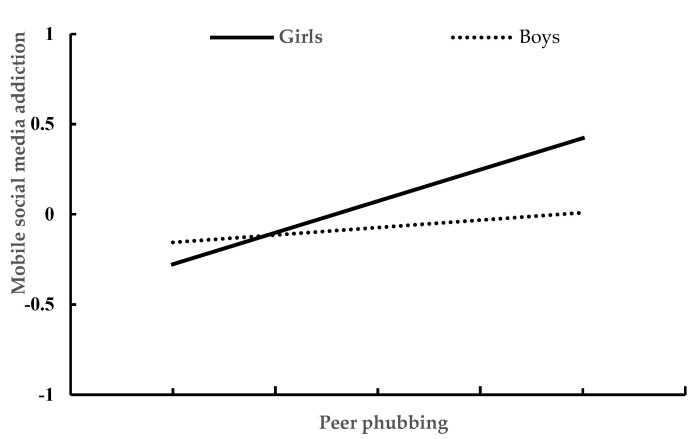
The relationship between peer phubbing and mobile social media addiction in girls and boys.

**Table 1 ijerph-19-10176-t001:** Test for the gender differences in the core variables.

Variables	Group	*M*	*SD*	*t*	*p*
Peer phubbing	Boys	2.728	0.895	−0.723	0.470
	Girls	2.773	0.917		
Loneliness	Boys	1.890	0.569	−4.411	<0.001
	Girls	2.089	0.734		
Mobile social media addiction	Boys	2.409	1.213	−2.415	<0.01
	Girls	2.707	1.295		

**Table 2 ijerph-19-10176-t002:** Intercorrelations between variables.

Variables	*M*	*SD*	1	2	3
1. Peer phubbing	2.753	0.907	—	0.343 ***	0.154 ***
2. Loneliness	1.999	0.671	0.548 ***	—	0.245 ***
3. Mobile social media addiction	2.571	1.267	0.508 ***	0.433 ***	—

Note. *** *p* < 0.001. Values above and below the diagonal represent girl and boy samples, respectively.

**Table 3 ijerph-19-10176-t003:** Mediation analysis of the role of loneliness.

Outcome Variables	Independent Variables	β	*SE*	*t*	*p*
Mobile social media addiction	Constant	0.001	0.032	0.001	0.999
	Age	−0.097 **	0.031	−3.154	<0.01
	Daily mobile phone use time	0.170 ***	0.037	4.575	<0.001
	Peer phubbing	0.339 ***	0.035	9.828	<0.001
Loneliness	Constant	0.001	0.031	0.001	0.999
	Age	−0.091 **	0.031	−2.954	<0.01
	Daily mobile phone use time	0.017	0.033	0.534	0.594
	Peer phubbing	0.453 ***	0.037	12.375	<0.001
Mobile social media addiction	Constant	0.001	0.031	0.001	0.999
	Age	−0.074 *	0.030	−2.483	<0.05
	Daily mobile phone use time	0.166 ***	0.035	4.684	<0.001
	Peer phubbing	0.225 ***	0.037	6.033	<0.001
	Loneliness	0.252 ***	0.039	6.533	<0.001

Note. N = 830. Bootstrap sample size = 5000. LL = low limit, CI = confidence interval, UL = upper limit. * *p* < 0.05. ** *p* < 0.01. *** *p* < 0.001.

**Table 4 ijerph-19-10176-t004:** Moderated mediation analysis.

Regression Models	β	*SE*	*t*	*p*
Mediator variable model for predicting loneliness				
Constant	−0.004	0.030	−0.125	0.901
Age	−0.078	0.031	−2.525	0.012
Daily mobile phone use time	0.015	0.031	0.488	0.626
Gender	0.256 ***	0.060	4.259	<0.001
Peer phubbing	0.448 ***	0.034	13.021	<0.001
Peer phubbing × Gender	0.300 ***	0.069	4.336	<0.001
Dependent variable model for predicting mobile social media addiction				
Constant	−0.004	0.031	−0.121	0.904
Age	−0.071	0.030	−2.380	0.018
Gender	0.147 *	0.064	2.302	<0.05
Peer phubbing	0.238 ***	0.035	6.732	<0.001
Loneliness	0.212 ***	0.040	5.293	<0.001
Peer phubbing × Gender	0.296 ***	0.066	4.465	<0.001
Conditional direct effect analysis at values of the moderator (gender)	β	Boot SE	BootLLCI	BootULCI
Boys	0.077	0.051	−0.023	0.176
Girls	0.373 ***	0.046	0.282	0.464
Conditional indirect effect analysis at values of the moderator (gender)	β	Boot SE	BootLLCI	BootULCI
Boys	0.060 **	0.015	0.035	0.095
Girls	0.124 ***	0.025	0.078	0.178

Note. N = 830. Bootstrap sample size = 5000. LL = low limit, CI = confidence interval, UL = upper limit. * *p* < 0.05. ** *p* < 0.01. *** *p* < 0.001.

## Data Availability

The data presented in this study are available on reasonable request from the corresponding author.

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
