# Peer review of "The Mediating Role of Loneliness and the Moderating Role of Gender between Peer Phubbing and Adolescent Mobile Social Media Addiction"

_ijerph, 2022, doi:10.3390/ijerph191610176_

Round 1
Reviewer 1 Report
ijerph-1814015
Peer phubbing and adolescent mobile social addiction: Testing the roles of loneliness and gender
The authors present results from a study looking at the problematic social media usage of 830 adolescents between 11 and 18 years of age. They investigate the phenomenon of peer phubbing in line with loneliness and social media addiction, an important area of study in this present day. The authors identified positive relationships between the investigated variables while noting the mediating role of loneliness and moderating role of gender. The study has a number of strengths and is written in a clear and detailed manner. Nevertheless, some minor revisions are suggested prior to publication:
Abstract
- It would be good to include the direction of the effect – whether peer phubbing and mobile social addiction are positively or negatively associated.
- More background information should be provided on peer phubbing in the abstract and why it is relevant to investigate
Introduction
- This point applies to the entire manuscript (including the abstract). The authors utilize the term “mobile social addiction” yet this is not an adequate term based on the field of literature and what they are studying. It appears as though the authors are looking at mobile social media addiction. Mobile social addiction is too broad and unclear a term and could involve any behaviors considered social on mobile. If the authors indeed mean to use the term more broadly, it should be defined.
- Prior to providing details of studies on the effects of social media addiction, I think it would be wise for the authors to define what they mean by addiction. What does this term account for that is not otherwise addressed by terms such as problematic social media use or compulsive social media use. As the authors utilize the term addiction repetitively and as a description of studies looking at Facebook and WeChat, it is important that they define their terms.
- Once again, the authors use the terms “too addicted to mobile phones” in the definition of peer phubbing. I would caution the overuse of the term “addicted” and perhaps use other more descriptive words. From my understanding, peer phubbing includes not only youth that are addicted to mobile phones but also ignoring someone in order to pay attention to your phone instead (whether they are addicted or not). These comments points to the importance of using clear language and descriptors and making sure their concepts are well defined.
Methods
- Good description of measures overall.
- I think the measure of mobile social addiction could use additional description of the items. Based on this section, I am still unsure what exactly the authors are measuring when it comes to “social addiction” if it is only social media, other apps… This is essential as this was also a point unclear to me in the introduction
- Statistics: the software used to conduct the analyses should be specified here – not only in the Results
- Were any assumption checks conducted by the authors.
Results
- The authors state: “The mediation effect of social anxiety accounted for 33.69% of the total effect. In other 145 words, loneliness partially mediated the association between peer pressure and mobile 146 social addiction.” This is the first time I read about social anxiety and peer pressure in this manuscript. Could the authors clarify this please? Were additional variables included or are the authors using different terms to describe them?
- The authors should specify “moderated mediation model” for section 3.3, not mediation model.
Discussion
- Typo page 7, line 201 “expends”
- I would remove all use of the words that suggest causation in the Discussion e.g., “cause”. As these results were reported from a cross-sectional study, the methodology utilized does not allow for any proof of causation to be established. For instance, you looked at whether loneliness is associated in mobile social addiction but mobile social addiction could also be associated with loneliness in the opposite direction. I would also add this as a limitation.
- I would add a discussion points on the reasons as to why phubbing was more strongly associated among females. Females also tend to be more likely to engage in indirect forms of bullying (ignoring, rejecting peers) and this may be a novel manifestation of this form of bullying.
- The authors name some practical implications and possible ways that schools, teachers and parents could help. However, there is a lack of specifying how exactly these institutions or people could help reduce these behaviors. These more specific suggestions should be added.
The conclusions are quite redundant and could be edited to be made more concise.
Author Response
#Review 1
ijerph-1814015
Peer phubbing and adolescent mobile social addiction: Testing the roles of loneliness and gender
The authors present results from a study looking at the problematic social media usage of 830 adolescents between 11 and 18 years of age. They investigate the phenomenon of peer phubbing in line with loneliness and social media addiction, an important area of study in this present day. The authors identified positive relationships between the investigated variables while noting the mediating role of loneliness and moderating role of gender. The study has a number of strengths and is written in a clear and detailed manner. Nevertheless, some minor revisions are suggested prior to publication:
Response: Thank you very much for your comments.
Abstract
It would be good to include the direction of the effect – whether peer phubbing and mobile social addiction are positively or negatively associated.
Response: According to your suggestion, we added some information to clarify the direction of the predictive effect of peer phubbing on mobile social addiction.
More background information should be provided on peer phubbing in the abstract and why it is relevant to investigate.
Response: According to your suggestion, we added some background information to highlight the severity of peer phubbing and the implications of exploring the effect of peer phubbing.
Introduction
This point applies to the entire manuscript (including the abstract). The authors utilize the term “mobile social addiction” yet this is not an adequate term based on the field of literature and what they are studying. It appears as though the authors are looking at mobile social media addiction. Mobile social addiction is too broad and unclear a term and could involve any behaviors considered social on mobile. If the authors indeed mean to use the term more broadly, it should be defined.
Response: Thank you very much for reminding us of this issue. We agree with your opinion that mobile social addiction is too broad and unclear a term and could involve any behaviors considered social on mobile. We aimed to explore mobile social media addiction. We modified the term “mobile social addiction” and used “mobile social media addiction” instead.
Prior to providing details of studies on the effects of social media addiction, I think it would be wise for the authors to define what they mean by addiction. What does this term account for that is not otherwise addressed by terms such as problematic social media use or compulsive social media use. As the authors utilize the term addiction repetitively and as a description of studies looking at Facebook and WeChat, it is important that they define their terms.
Response: Thank you very much for highlighting this issue. According to your suggestion, we define the term “mobile social media addiction” in more detail in the manuscript.
Once again, the authors use the terms “too addicted to mobile phones” in the definition of peer phubbing. I would caution the overuse of the term “addicted” and perhaps use other more descriptive words. From my understanding, peer phubbing includes not only youth that are addicted to mobile phones but also ignoring someone in order to pay attention to your phone instead (whether they are addicted or not). These comments points to the importance of using clear language and descriptors and making sure their concepts are well defined.
Response: Thank you very much for your detailed comments. We have revised the terms “too addicted to mobile phones.” We highlighted more the meaning of ignoring someone in order to pay attention to mobile phones. We also checked other potential problems in using clear language and descriptors.
Methods
Good description of measures overall.
I think the measure of mobile social addiction could use additional description of the items. Based on this section, I am still unsure what exactly the authors are measuring when it comes to “social addiction” if it is only social media, other apps… This is essential as this was also a point unclear to me in the introduction.
Response: According to your suggestion, we have added more descriptions of the items. We also used the term “mobile social media addiction” instead of “mobile social addiction” throughout the manuscript.
Statistics: the software used to conduct the analyses should be specified here – not only in the Results
Response: We have specified the software used in the Statistics section.
Were any assumption checks conducted by the authors.
Response: We added that we conducted a t-test to examine the gender differences in the scores of peer phubbing, loneliness, and mobile social media addiction.
Results
The authors state: “The mediation effect of social anxiety accounted for 33.69% of the total effect. In other 145 words, loneliness partially mediated the association between peer pressure and mobile social addiction.” This is the first time I read about social anxiety and peer pressure in this manuscript. Could the authors clarify this please? Were additional variables included or are the authors using different terms to describe them?
Response: Sorry for the mistake in writing. No additional variables were included, and no different terms were used. We have deleted the word “social anxiety” and “pressure” and used “loneliness” and “phubbing” instead, respectively.
The authors should specify “moderated mediation model” for section 3.3, not mediation model.
Response: We added the word “moderated” and revised the “mediation model” in section 3.3 into “moderated mediation model.”
Discussion
Typo page 7, line 201 “expends”
Response: Sorry for the mistake in writing. We have revised the word “expends” into “expands.”
I would remove all use of the words that suggest causation in the Discussion e.g., “cause”. As these results were reported from a cross-sectional study, the methodology utilized does not allow for any proof of causation to be established. For instance, you looked at whether loneliness is associated in mobile social addiction but mobile social addiction could also be associated with loneliness in the opposite direction. I would also add this as a limitation.
Response: Thank you very much for highlighting this issue. We have removed all use of the words that suggest causation, such as “cause” and “increase.” The cross-sectional design we used cannot strictly confirm causal relationships. We have added this as a limitation in the Discussion section of the original manuscript.
I would add a discussion points on the reasons as to why phubbing was more strongly associated among females. Females also tend to be more likely to engage in indirect forms of bullying (ignoring, rejecting peers) and this may be a novel manifestation of this form of bullying.
Response: We agree with your opinion that phubbing may be a novel manifestation of indirect forms of bullying victimization. We have added this to discuss the reasons as to why phubbing was more strongly associated among females.
The authors name some practical implications and possible ways that schools, teachers and parents could help. However, there is a lack of specifying how exactly these institutions or people could help reduce these behaviors. These more specific suggestions should be added.
Response: Thank you very much for highlighting this issue. According to your this comment, we have added some information to specify how institutions or people could help reduce peer phubbing, loneliness, and mobile social media addiction.
The conclusions are quite redundant and could be edited to be made more concise.
Response: According to your suggestion, we shortened the conclusions to make this section more concise.
Reviewer 2 Report
[1] First, I add some line-based comments:
1. line 13-15: odd use of phrase "in the association"; suggestion to "The present study examined the mediation of loneliness between peer phubbing and mobile social addiction among..." (i.e. remove the phrase 'in the association' entirely)
2. line 29: cite that differentiation better with more descriptions who invented this concept. This is important given there is no current standard of definition even of 'mobile addiction' at all. There are only two currently accepted behavioral (non-chemical) addictions in the DSM-V: gambling disorder and internet gaming disorder. Discussion is currently raging over whether to add a category for 'mobile addiction' and how to define it though. Relate the paper as a contribution to defining effects of 'mobile social addiction' as a subtype? Mobile phones have begun to be considered though it is hardly clear that mobile phones are causes of 'addiction' themselves per se. So use this term 'mobile addiction' and 'mobile social addiction' with caveats stated about all this that the 'phone' itself may not be the core problem itself at all.
3. line 68-71: I am kind of confused by the statement of their independent variables (IV) and dependent variables (DV) proposed. They are saying the presumed/hypothesized causality of peer phubbing is the IV upon the DV of loneliness, yet then they want to say that loneliness is itself an IV upon 'addiction'. So it seems that phubbing is not the only IV here by their own discussion, yet they do not mention loneliness as IV as well upon 'mobile social addiction.' So I think their model discussion could be summarized a bit better for this dual DV/IV of lonleness as a mediator. And if it is a mediator, can it really be a DV of phubbing at all, when lonliness is the direct connection that is more important in their mind? Plus, as they say in the conclusion, this is non-time-series data so unable to really discuss causality per se, and can only provide some interesting correlations, so I wonder if a model 'testing causality and mediation' can be done with non-time-series data at all?
4. line 83: should state earlier that they have two moderating hypotheses about loneliness and about gender both equally separate on 'mobile social addiction', correct? Or are they saying that there is only one moderator on 'mobile social addiction', which is loneliness, which itself is moderated by gender? The question is are both of these equal separate factors on 'mobile social addiction' or is their second moderator of gender just an ancillary moderator on their first moderator of loneliness? What are they testing? I think they are testing both pathways, right?
5. line 114-15: rephrase the English translation of this question to avoid the double negative ["I canNOT stand NOT looking"] in the sentence, to achieve proper English grammar. "I cannot stand being unable to look...'.
6. line 130: says "Test for homogeneous between the mindfulness group and control group" I think they mean "homogeneity?" Plus for the rest of the sentence, this is the first time they mention 'mindfulness group' or 'control group' in the whole paper. Their model (line 96-98) does not describe any unique 'mindfulness group or any unique 'control group' at all, only the basic 830 teens tested. Please clear this up what they mean by 'mindfulness group' or 'control group' where there were nothing of the kind as a variable? Or was there such a separate group that they are not analyzing?
7. Split Table 1 in a better way, to put boy and girl groups of mobile social addiction on the same page, this one or the next, instead of split over two pages.
8. line 94: China was misspelled as "Chine".
General comments:
1. It is good their measurements have great Cronbach's alpha's, yet if you look at their standard deviations (SD) in Table 1, then BOTH boys and girls have very wide SDs which indicates that 'gender' is hardly a clear strong 'variable' between boys and girls, so it cannot have a clear moderating effect itself, though is a far more individualized psychological issue to a particular person. For instance, 'peer phubbing' and 'mobile social addiction' both have similar and huge SD's for boys AND girls, and boys only have a small tighter distribution in their male loneliness. This should affect discussion in the analysis. I think it should, though they carry on as if gender can be treated as some major predicting variable, when it clearly is all over the place with boys and girls equally in the data. Only the averages are different, though truly, it is not that large of a difference in averages either in Table 1. So the people they are analyzing are not really all that different by gender, given these wide SDs in these two variables? Comment or address that?
2. Moreover, in Table 1, the measure for 'peer phubbing' itself is without any statistical significance, right? (in the last column?, 0.470 as the p (value)?) I am confused how they can base their whole analysis off an IV of peer phubbing when there is a lack of statistical significance in peer phubbing on some level here as a measure, right?
3. Plus in Table 1, the means of peer phubbing by boys and girls do not seem different by gender 2.728 and 2.773 at all, respectively, yet they then do regression analysis on it and claim gender is really important in peer phubbing, yet the means themselves are virtually the same for gender? I hypotheize they are fixating on their statistical models instead of really noting the data itself shows almost no connection/variation by gender at all as a moderating effect upon peer phubbing in these correlations otherwise the means would be very different.
4. on line 143-44, it is said "The bias-corrected percentile bootstrap method showed that the mediating effect of loneliness was 0.116...." This is a very low correlation, so it is hard to justify a whole article/discussion off of any important effects of this lonelinessas a mediating factor at all? Comment? It is a VERY tiny effect, this mediating effect of loneliness. This means an outsized analysis of loneliness as a mediating effect may be completely misplaced attention on multiple other factors per person instead of any 'easy answers' here It would seem that the model is confirmed, though it is not very statistically important a moderator at all upon 'mobile social addiction', which negates their whole line of discussion. I suggest you conclude with something like this?
line 145-47: quoting: "In other words, loneliness partially mediated the association between peer pressure and mobile 146 social addiction." I think it would be better to say "In other words, loneliness only minimally mediated any association between peer pressure and mobile social addiction."
line 161-63: quoting "The association between peer phubbing and mobile social addiction was strong in girls (β = 0.373, p < 0.001), but it was not significant in boys (β 162 = 0.007, p > 0.001)." Do you agree it is more accurate to describe your results as: "The association between peer phubbing and mobile social addiction was only moderate in girls (β = 0.373, p < 0.001), and almost meaningless in boys (β = 0.007, p > 0.001), yet both findings were statistically significant in terms of p-value."
line 171-73: Once more quoting something here, they overstate the importance of loneliness as a moderator: They say "For girls, peer phubbing had a strong effect on mobile social addiction through the mediating effect of loneliness (β = 0.124, p < 0.01). For boys, however, the mediating effect of loneliness was relatively low (β = 0.06, p < 0.01)." I think it is more accuate to say : "For girls, peer phubbing had a very small effect on mobile social addiction through the mediating effect of loneliness (β = 0.124, p < 0.01). For boys, the mediating effect of loneliness was almost zero (β = 0.06, p < 0.01). This connects to line 161-63 better as well as the true finding of the study: a minor moderating effect of phubbing on loneliness via girls, and almost zero effects of phubbing on loneliness in boys. Comment?
in Table 4, phubbing itself shows a far stronger effect on loneliness than gender on loneliness (even though they talk more about gender), and phubbing itself shows a far stronger effect on mobile social addiction than gender (even though they talk more about gender). The low and seemingly unimportant gender moderation issue on both phubbing and mobile social addiction goes well with the above two points in this commentary: that form some motivation that they want to overstate a gender element here when it is really very tiny in girls and non-existent in boys.
line 184-85: quoting: "The direct effect of peer phubbing and the indirect effect through loneliness were stronger in girls than in boys." Is it better to say that "The direct effect of peer phubbing and the indirect effect through loneliness 184 were very minor in girls and almost non-existent in boys." There is nothing "strong" about gender mediation findings at all, though they keep using these words like 'strong,' in a relative sense, between minor and near zero effects.
line 195: quoting "We found that loneliness partially mediated." I think you found that "loneliness only minimally mediated the association between female peer phubbing and female adolescent mobile social addiction, and had almost zero mediation potential on the association between male peer phubbing and male adolescent mobile social addiction."
line 223: quoting "In line with these results, girls in our study have higher levels of mobile social addiction than males." I would say "In line with these results, girls in our study (looking at Table 1) have only very minimally higher levels of mobile social addiction, loneliness, and mobile social addiction than males."
In short, in one comment: I think the authors are overstating a 'gender variable' finding/narrative on mediating various issues, tremendously.
Author Response
#Reviewer 2
[1] First, I add some line-based comments:
Response: Thank you very much for your comments.
- line 13-15: odd use of phrase "in the association"; suggestion to "The present study examined the mediation of loneliness between peer phubbing and mobile social addiction among..." (i.e. remove the phrase 'in the association' entirely)
Response: According to your suggestion, we have removed the phrase “in the association.”
- line 29: cite that differentiation better with more descriptions who invented this concept. This is important given there is no current standard of definition even of 'mobile addiction' at all. There are only two currently accepted behavioral (non-chemical) addictions in the DSM-V: gambling disorder and internet gaming disorder. Discussion is currently raging over whether to add a category for 'mobile addiction' and how to define it though. Relate the paper as a contribution to defining effects of 'mobile social addiction' as a subtype? Mobile phones have begun to be considered though it is hardly clear that mobile phones are causes of 'addiction' themselves per se. So use this term 'mobile addiction' and 'mobile social addiction' with caveats stated about all this that the 'phone' itself may not be the core problem itself at all.
Response: Thank you very much for highlighting this issue. We agree with your opinion that discussion is currently raging over whether to add a category for “mobile addiction” and how to define it though. We also acknowledged that the mobile phone itself might not be the core problem of addiction at all. Some researchers also used the term mobile phone dependence or problematic mobile phone use. Although different researchers use different terms, they seem to all agree with the seriousness of overusing mobile phones (Liu et al., 2022). Different from other terms such as problematic mobile phone use or compulsive mobile phone use, mobile phone addiction was used to highlight that behaviors of overusing mobile phones have four typical symptoms of mobile phone addiction (i.e., loss of control, anxiety and feeling lost, withdrawal and escape, and productivity loss). According to the comments of Review 1, we used the term “mobile social media addiction” throughout the manuscript. We cited more descriptions of “social media addiction” and “mobile social media addiction”.
Liu, Q., Zhou, Z., & Eichenberg, C. (2022). Significant Influencing Factors and Effective Interventions of Mobile Phone Addiction. Frontiers in Psychology, 13. https://doi.org/10.3389/fpsyg.2022.909444
- line 68-71: I am kind of confused by the statement of their independent variables (IV) and dependent variables (DV) proposed.They are saying the presumed/hypothesized causality of peer phubbing is the IV upon the DV of loneliness, yet then they want to say that loneliness is itself an IV upon 'addiction'. So it seems that phubbing is not the only IV here by their own discussion, yet they do not mention loneliness as IV as well upon 'mobile social addiction.' So I think their model discussion could be summarized a bit better for this dual DV/IV of loneliness as a mediator. And if it is a mediator, can it really be a DV of phubbing at all, when loneliness is the direct connection that is more important in their mind? Plus, as they say in the conclusion, this is non-time-series data so unable to really discuss causality per se, and can only provide some interesting correlations, so I wonder if a model 'testing causality and mediation' can be done with non-time-series data at all?
Response: Thank you very much for reminding us of this issue. We hypothesized that loneliness is a mediator linking peer phubbing to mobile social media addiction. Thus, loneliness is the dependent variable of peer phubbing and also the independent variable of mobile social media addiction. We provided some information on the effect of loneliness on mobile social addiction in the third paragraph of the Introduction section of the original manuscript. According to your suggestion, we added more statements to highlight the effect of loneliness on mobile social media addiction. In addition, similar to many studies exploring the mediation model through a cross-sectional design, our study has the shortcoming that it can not strictly confirm the causal relationships. Although some researchers believe that causal relationships can not be determined only from the data and previous theoretical and empirical evidence should also be considered, the defects in the data still need to be taken seriously. We included this as a limitation in the Discussion section. In future research, we would try to adopt longitudinal research more likely to demonstrate the causal relationships.
- line 83: should state earlier that they have two moderating hypotheses about loneliness and about gender both equally separate on 'mobile social addiction', correct? Or are they saying that there is only one moderator on 'mobile social addiction', which is loneliness, which itself is moderated by gender? The question is are both of these equal separate factors on 'mobile social addiction' or is their second moderator of gender just an ancillary moderator on their first moderator of loneliness? What are they testing? I think they are testing both pathways, right?
Response: We tested the roles of loneliness and gender between peer phubbing and mobile social media addiction. Loneliness was hypothesized as a mediator (not a moderator), and gender was a moderator. We did not treat both of them as moderators. Loneliness acted as a mediator linking peer phubbing to mobile social addiction. In other words, peer phubbing was positively associated with loneliness, which, in turn, was positively associated with mobile social media addiction. Gender acted as a moderator that may influence the direct effect of peer phubbing on mobile social addiction and the mediation of loneliness. In other words, peer phubbing did not influence the moderator (gender), but the direct and indirect effects of peer phubbing would be different in adolescents of different genders.
- line 114-15: rephrase the English translation of this question to avoid the double negative ["I canNOT stand NOT looking"] in the sentence, to achieve proper English grammar. "I cannot stand being unable to look...'.
Response: According to your comment, we have revised the sample item of the Mobile Social Addiction Scale. We used another two sample items that did not include the double negative meaning in the sentence.
- line 130: says "Test for homogeneous between the mindfulness group and control group" I think they mean "homogeneity?" Plus for the rest of the sentence, this is the first time they mention 'mindfulness group' or 'control group' in the whole paper. Their model (line 96-98) does not describe any unique 'mindfulness group or any unique 'control group' at all, only the basic 830 teens tested. Please clear this up what they mean by 'mindfulness group' or 'control group' where there were nothing of the kind as a variable? Or was there such a separate group that they are not analyzing?
Response: Sorry for the mistake in writing. There is no mindfulness group or control group in our study. The title of Table 1 used the title of another table in our previous research by mistake. We have checked all the titles of the tables in the manuscript.
- Split Table 1 in a better way, to put boy and girl groups of mobile social addiction on the same page, this one or the next, instead of split over two pages.
Response: We have modified Table 1 to put boy and girl groups of mobile social addiction on the same page.
- line 94: China was misspelled as "Chine".
Response: Sorry for the mistake in writing. Thank you for your careful reading. We have revised this mistake.
General comments:
- It is good their measurements have great Cronbach's alpha's, yet if you look at their standard deviations (SD) in Table 1, then BOTH boys and girls have very wide SDs which indicates that 'gender' is hardly a clear strong 'variable' between boys and girls, so it cannot have a clear moderating effect itself, though is a far more individualized psychological issue to a particular person. For instance, 'peer phubbing' and 'mobile social addiction' both have similar and huge SD's for boys AND girls, and boys only have a small tighter distribution in their male loneliness. This should affect discussion in the analysis. I think it should, though they carry on as if gender can be treated as some major predicting variable, when it clearly is all over the place with boys and girls equally in the data. Only the averages are different, though truly, it is not that large of a difference in averages either in Table 1. So the people they are analyzing are not really all that different by gender, given these wide SDs in these two variables? Comment or address that?
Response: Thank you very much for your comments. We tested the gender differences in the direct relationship between peer phubbing and mobile social addiction and the indirect through loneliness. We did not focus too much on gender differences in peer phubbing, loneliness, and mobile social addiction themselves. Statistically, gender differences in the relationships between some variables were not determined by gender differences in the variables themselves. The moderating effect was generated by the interaction of the independent variable and the moderator variable through the mean center for products, not the moderator variable itself. Some studies showing no significant gender differences in the core variables (or no significant effect of gender on the core variables) confirmed that gender could still act as a moderator between the independent variable and the dependent variable (e.g., Cabello, Gutiérrez-Cobo, & Fernandez-Berrocal, 2017; Ding et al., 2017; Iorfa et al., 2020; Kacmar et al., 2011; Wang et al., 2021). Therefore, the differences in mean values and SDs in the variables may not have significant impact on analyzing the gender differences in the direct and indirect effects of peer phubbing.
Cabello, R., Gutiérrez-Cobo, M. J., & Fernandez-Berrocal, P. (2017). Parental education and aggressive behavior in children: A moderated-mediation model for inhibitory control and gender. Frontiers in Psychology, 8, 1181. https://doi.org/10.3389/fpsyg.2017.01181
Ding, Q., Zhang, Y. X., Wei, H., Huang, F., & Zhou, Z. K. (2017). Passive social network site use and subjective well-being among Chinese university students: A moderated mediation model of envy and gender. Personality and Individual Differences, 113, 142–146.
Iorfa, S. K., Ottu, I. F., Oguntayo, R., Ayandele, O., Kolawole, S. O., Gandi, J. C., ... & Olapegba, P. O. (2020). COVID-19 knowledge, risk perception, and precautionary behavior among Nigerians: A moderated mediation approach. Frontiers in Psychology, 11, 566773. https://doi.org/10.3389/fpsyg.2020.566773
Kacmar, K. M., Bachrach, D. G., Harris, K. J., & Zivnuska, S. (2011). Fostering good citizenship through ethical leadership: Exploring the moderating role of gender and organizational politics. Journal of Applied Psychology, 96(3), 633–642.
Wang, W., Qian, G., Wang, X., Lei, L., Hu, Q., Chen, J., & Jiang, S. (2021). Mobile social media use and self-identity among Chinese adolescents: The mediating effect of friendship quality and the moderating role of gender. Current Psychology, 40(9), 4479–4487.
- Moreover, in Table 1, the measure for 'peer phubbing' itself is without any statistical significance, right? (in the last column?, 0.470 as the p (value)?) I am confused how they can base their whole analysis off an IV of peer phubbing when there is a lack of statistical significance in peer phubbing on some level here as a measure, right?
Response: Statistically, gender differences in the relationships between some variables were not determined by gender differences in the variables themselves, in particular the gender differences in the independent variable. The moderating effect was generated by the interaction of the independent variable (peer phubbing) and the moderator variable (gender) through the mean center for products, not the independent variable (peer phubbing) or the moderator variable (gender) itself. Some studies showing no significant gender differences in the core variables (or no significant effect of gender on the core variables) confirmed that gender could still act as a moderator between the independent variable and the dependent variable (e.g., Cabello, Gutiérrez-Cobo, & Fernandez-Berrocal, 2017; Ding et al., 2017; Iorfa et al., 2020; Kacmar et al., 2011; Wang et al., 2021). Thus, although there were no significant gender differences in peer phubbing, the effect of peer phubbing may still show significant gender differences. In a moderation model, individuals with the same independent variable level show different levels of dependent variables due to the differences in the moderator variable. Therefore, differences in the independent variable are not necessary for analyzing a moderation model.
- Plus in Table 1, the means of peer phubbing by boys and girls do not seem different by gender 2.728 and 2.773 at all, respectively, yet they then do regression analysis on it and claim gender is really important in peer phubbing, yet the means themselves are virtually the same for gender? I hypotheize they are fixating on their statistical models instead of really noting the data itself shows almost no connection/variation by gender at all as a moderating effect upon peer phubbing in these correlations otherwise the means would be very different.
Response: We did not claim gender differences in peer phubbing. We reported that no significant gender differences wore found in the scores of peer phubbing in the original manuscript. We have to highlight again that gender differences in the independent variable are not a necessary prerequisite for analyzing the gender differences in the effect of the independent variable. In a moderation model, individuals with the same independent variable level show different levels of dependent variables due to the differences in the moderator variable. These individuals do not need to have significant differences in the levels of the independent variable. Statistically, gender differences in the relationships between some variables were not determined by gender differences in the variables themselves. Some studies showing no significant gender differences in the core variables (or no significant effect of gender on the core variables) confirmed that gender could still act as a moderator between the independent variable and the dependent variable (e.g., Cabello, Gutiérrez-Cobo, & Fernandez-Berrocal, 2017; Ding et al., 2017; Iorfa et al., 2020; Kacmar et al., 2011; Wang et al., 2021).
- on line 143-44, it is said "The bias-corrected percentile bootstrap method showed that the mediating effect of loneliness was 0.116...." This is a very low correlation, so it is hard to justify a whole article/discussion off of any important effects of this lonelinessas a mediating factor at all? Comment? It is a VERY tiny effect, this mediating effect of loneliness. This means an outsized analysis of loneliness as a mediating effect may be completely misplaced attention on multiple other factors per person instead of any 'easy answers' here It would seem that the model is confirmed, though it is not very statistically important a moderator at all upon 'mobile social addiction', which negates their whole line of discussion. I suggest you conclude with something like this?
Response: Thank you very much for reminding us of this issue. The mediating effect value is not a correlation coefficient. It is the product of the independent and dependent variables. Moreover, the mediating effect values are usually not too high in the questionnaire research. Some studies reported a mediating effect below 0.10 but still highlighted that the mediator played an important role in linking the independent variable to the dependent variable (e.g., Kim, Kwak, & Kim, 2022; Sela et al., 2020; Sun, Liu, & Yu, 2019; Xie, Chen, Zhu, & He, 2019; Yan, Lin, Su, & Liu, 2018). In addition, a dependent variable is usually influenced by multiple factors. A certain independent variable and its mediator variable generally play only a part in explaining the levels of the dependent variable. In this case, the effect size of the mediating effect is not only determined by the absolute values of the effect but also by the ratio of the mediating effect on the total effect of the independent variable. In our study, the ratio of the mediating effect on the total effect of peer phubbing is 33.69, which may be considered a moderate effect size because the ratio above 60% often causes a complete mediation model in which the independent variable is associated with the dependent variable completely through the mediator.
We agree with your opinion that other factors may also mediate between peer phubbing and mobile social media addiction because loneliness only partially mediated the relationship between peer phubbing and mobile social media addiction. The necessity of exploring other mediators had been included in the Discussion section.
Kim, C., Kwak, K., & Kim, Y. (2022). The relationship between stress and smartphone addiction among adolescents: the mediating effect of grit. Current Psychology, 1–9. https://doi.org/10.1007/s12144-022-03367-6
Sela, Y., Zach, M., Amichay-Hamburger, Y., Mishali, M., & Omer, H. (2020). Family environment and problematic internet use among adolescents: The mediating roles of depression and fear of missing out. Computers in Human Behavior, 106, 106226.
Sun, J., Liu, Q., & Yu, S. (2019). Child neglect, psychological abuse and smartphone addiction among Chinese adolescents: The roles of emotional intelligence and coping style. Computers in Human Behavior, 90, 74–83.
Xie, X., Chen, W., Zhu, X., & He, D. (2019). Parents' phubbing increases Adolescents' Mobile phone addiction: Roles of parent-child attachment, deviant peers, and gender. Children and Youth Services Review, 105, 104426.
Yan, Y. W., Lin, R. M., Su, Y. K., & Liu, M. Y. (2018). The relationship between adolescent academic stress and sleep quality: A multiple mediation model. Social Behavior and Personality: An International Journal, 46(1), 63–77.
line 145-47: quoting: "In other words, loneliness partially mediated the association between peer pressure and mobile social addiction." I think it would be better to say "In other words, loneliness only minimally mediated any association between peer pressure and mobile social addiction."
Response: The term “peer pressure” is a mistake in writing. It should be peer phubbing. According to this comment, we revised this sentence as “In other words, loneliness played a weak but significant mediating effect in the relationship between peer phubbing and mobile social media addiction.”
line 161-63: quoting "The association between peer phubbing and mobile social addiction was strong in girls (β = 0.373, p < 0.001), but it was not significant in boys (β = 0.007, p > 0.001)." Do you agree it is more accurate to describe your results as: "The association between peer phubbing and mobile social addiction was only moderate in girls (β = 0.373, p < 0.001), and almost meaningless in boys (β = 0.007, p > 0.001), yet both findings were statistically significant in terms of p-value."
Response: Thank you for your detailed suggestion. According to your suggestion, we revised the mentioned sentence as “The association between peer phubbing and mobile social addiction was moderate in girls (β = 0.373, p < 0.001), but almost meaningless in boys (β = 0.007, p > 0.05).” We deleted the sentence “yet both findings were statistically significant in terms of p-value” because the effect of peer phubbing on mobile social addiction in boys is not significant (p > 0.05). In the original manuscript, there is a mistake in the p-value of the effect in boys (p > 0.001). The p-value there should be p > 0.05.
line 171-73: Once more quoting something here, they overstate the importance of loneliness as a moderator: They say "For girls, peer phubbing had a strong effect on mobile social addiction through the mediating effect of loneliness (β = 0.124, p < 0.01). For boys, however, the mediating effect of loneliness was relatively low (β = 0.06, p < 0.01)." I think it is more accurate to say: "For girls, peer phubbing had a very small effect on mobile social addiction through the mediating effect of loneliness (β = 0.124, p < 0.01). For boys, the mediating effect of loneliness was almost zero (β = 0.06, p < 0.01). This connects to line 161-63 better as well as the true finding of the study: a minor moderating effect of phubbing on loneliness via girls, and almost zero effects of phubbing on loneliness in boys. Comment?
Response: Thank you for your suggestion. However, we cannot use the sentence you provided because the mediating effect of loneliness in boys was significant (p < 0.01), although the effect is low. Statistically, the small but significant effect should not be considered as almost zero effect. Some studies reported a mediating effect below 0.10 but still highlighted that the mediator played an important role in linking the independent variable to the dependent variable (e.g., Kim, Kwak, & Kim, 2022; Sela et al., 2020; Sun, Liu, & Yu, 2019; Xie, Chen, Zhu, & He, 2019; Yan, Lin, Su, & Liu, 2018). We revised the above sentence to highlight that the mediating effect of loneliness between peer phubbing and mobile social media addiction is relatively higher in girls than in boys but did not highlight the effect values.
in Table 4, phubbing itself shows a far stronger effect on loneliness than gender on loneliness (even though they talk more about gender), and phubbing itself shows a far stronger effect on mobile social addiction than gender (even though they talk more about gender). The low and seemingly unimportant gender moderation issue on both phubbing and mobile social addiction goes well with the above two points in this commentary: that form some motivation that they want to overstate a gender element here when it is really very tiny in girls and non-existent in boys.
Response: We focused on gender differences in the direct and indirect effects of peer phubbing on mobile social media addiction, but not gender differences in mobile social addiction or the effect of gender on loneliness or mobile phone addiction. These were different issues. Statistically, gender differences in mobile social addiction or the effect of gender on loneliness or mobile phone addiction did not determine the moderating effect of gender in the direct and indirect effects of peer phubbing on mobile social media addiction. The moderating effect was generated by the interaction of the independent variable (peer phubbing) and the moderator variable (gender) through the mean center for products, not the moderator variable (gender) itself. The low or high effect of the moderator variable itself is not equal to the effect of the interaction. The effect value of the interaction is 0.296 (see Table 4), which should not be considered low, given that the moderating effect values are usually not too high in questionnaire research.
line 184-85: quoting: "The direct effect of peer phubbing and the indirect effect through loneliness were stronger in girls than in boys." Is it better to say that "The direct effect of peer phubbing and the indirect effect through loneliness were very minor in girls and almost non-existent in boys." There is nothing "strong" about gender mediation findings at all, though they keep using these words like 'strong,' in a relative sense, between minor and near zero effects.
Response: Thank you very much for your suggestion. The direct effect of peer phubbing was significant in girls but not significant in boys. The indirect effect of loneliness was significant in both girls and boys. Again, we have to highlight that the small but significant effect should not be considered as almost zero effect. As you said, we used the word “strong” in a relative sense but not an absolute sense. In the relative sense, the direct and indirect effects were stronger in girls than in boys. We deleted the words “strong” and “stronger” according to your suggestion. We revised the mentioned sentence as “The direct effect of peer phubbing and the indirect effect through loneliness were relatively higher in girls than in boys.”
line 195: quoting "We found that loneliness partially mediated." I think you found that "loneliness only minimally mediated the association between female peer phubbing and female adolescent mobile social addiction, and had almost zero mediation potential on the association between male peer phubbing and male adolescent mobile social addiction."
Response: Thank you very much for your suggestion. However, we cannot use the sentence you provided because our sentence was used to show the general situation in all participants. In general, loneliness indeed partially mediated peer phubbing and mobile social media addiction. We specified the mediating effect of loneliness in boys and girls in the fourth paragraph of the Discussion section in the original manuscript.
Moreover, the small but statistically significant effect should not be considered as almost zero effect.
line 223: quoting "In line with these results, girls in our study have higher levels of mobile social addiction than males." I would say "In line with these results, girls in our study (looking at Table 1) have only very minimally higher levels of mobile social addiction, loneliness, and mobile social addiction than males."
Response: According to your suggestion, we revised the mentioned sentence as “In line with these results, girls in our study have relatively higher levels of loneliness and mobile social addiction than males.” We highlighted that the word “higher” is in a relative sense but not an absolute sense.
In short, in one comment: I think the authors are overstating a 'gender variable' finding/narrative on mediating various issues, tremendously.
Response: We made some revisions according to your suggestions. However, after referring to many questionnaire studies on the moderated mediation model, we keep some sentences in the original manuscript. The moderating role of gender in the mediation model is not equal to the gender differences in the variables themselves. Compared with many other previous studies, the mediating effect and moderating effect in our study were even slightly higher.
We thank you very much for your careful reading and constructive comments.
Round 2
Reviewer 2 Report
Six suggested adjustments:
1. use the idea/word "hypothesis" instead of the general word "question". You are testing hypotheses instead of just answering questions. And define more clearly what you mean by mediation and moderation.
this applies to the zone around line 114:
so, my suggestion is for this section to read:
"We aimed to address two main hypotheses: (a) does loneliness act as a mediator linking peer-phubbing to mobile social media addiction in general for all groups, and in a disaggregated sense, (b) does gender act as a moderator regulating direct and indirect effects upon a gender's peer phubbing. [Then, define here what you mean by the differentiation of 'mediator' and 'moderator', for the reader.] The results would help uncover if peer phubbing is associated with adolescent mobile phone addiction and whether this applies to all people or to only certain gender groups in which the effect of peer phubbing may be potent.
2.
line 185 says "the SPSS 23.0"
suggest it be this phrase:
"the statistical software package SPSS 23.0"
[English syntax issue: 'the' is put on a general word, instead of such a specific noun like SPSS 23.0, just like you would say "the cat" instead of 'the Molly"]
3.
line 365:
you wrote: "Females are also more likely to engage in indirect forms 367 of bullying (e.g., ignoring, rejecting, or relational aggression) victimization."
This is interesting though you should add a citation for this, otherwise if it is just another hypothesis, just say:
"Further research may test if females as a group (more than males) are more likely to engage in such indirect forms of bullying victimization (e.g., ignoring, rejecting, or relational aggression), as we hypothesize female peer phubbing may be a fresh manifestation of an older common form of bullying victimization perhaps seen more in females than males."
4.
line 439:
I feel you still overstate the findings in the conclusions
you wrote: "in that these two effects were more potent in girls than in boys."
suggestion: "in that these two effects were slightly more potent in girls and almost nonexistent in boys." Correct? If not, explain in a bit more detail.
Plus, I would like to see a sentence somewhere (either in the conclusions or in the earlier caveat section) saying that given these weak categorical associations (of male and female), that other factors on the macro-level of the school organizational environment or the micro-level of individual psychology may be a better route to helping people, with either school based changes or with individually-tailored psychological consulting and individually tailored interventions, or both, instead of presuming you have to have only centralized managerial and categorical/sterotypical based interventions where you divide people in to male and female to give then the 'only two' (sic) treatments.
5.
line 439:
from this: "The findings highlight how peer phubbing is directly and indirectly associated with adolescent mobile social media addiction and for whom the direct and indirect effects of peer phubbing are potent."
to this suggestion:
"The findings highlight how peer phubbing is directly and indirectly associated with more female adolescent mobile social media addiction, and how female direct and indirect effects of peer phubbing are more potent on females than on boys, when comparing averages of both gender groups."
Correct? If not, explain a bit more, and you can of course adjust the suggested wording.
6.
suggested title adjustment
from this:
"Peer Phubbing and Adolescent Mobile Social Media Addiction: Testing the Roles [what roles?] of Loneliness and Gender"
to the suggestion:
Do Loneliness and Gender Act as Moderators Between Peer Phubbing and Adolescent Mobile Social Media Addiction?
What do you think? I just think the old title could be more succinct and clear of what is being tested since in the suggested version, the discussion of the IV as a mediator comes first, then using a verb on the DV, instead of a colon splitting the title into two indirect parts and leaving the 'role' between the two phrases quite undefined. Adjust using your own ideas or use this idea, yet please, adjust it somehow, I suggest.
Author Response
Response to Comments
#Reviewer 2
Six suggested adjustments:
- use the idea/word "hypothesis" instead of the general word "question". You are testing hypotheses instead of just answering questions. And define more clearly what you mean by mediation and moderation.
this applies to the zone around line 114:
so, my suggestion is for this section to read:
"We aimed to address two main hypotheses: (a) does loneliness act as a mediator linking peer-phubbing to mobile social media addiction in general for all groups, and in a disaggregated sense, (b) does gender act as a moderator regulating direct and indirect effects upon a gender's peer phubbing. [Then, define here what you mean by the differentiation of 'mediator' and 'moderator', for the reader.] The results would help uncover if peer phubbing is associated with adolescent mobile phone addiction and whether this applies to all people or to only certain gender groups in which the effect of peer phubbing may be potent.
Response: Thank you very much for this suggestion. We used the idea/word "hypothesis" instead of the general word "question." We also defined more clearly what we mean by mediation and moderation.
2.
line 185 says "the SPSS 23.0"
suggest it be this phrase:
"the statistical software package SPSS 23.0"
[English syntax issue: 'the' is put on a general word, instead of such a specific noun like SPSS 23.0, just like you would say "the cat" instead of 'the Molly"]
Response: Thank you very much for reminding us of this issue. We have changed “the SPSS 23.0” to “the statistical software package SPSS 23.0.”
3.
line 365:
you wrote: "Females are also more likely to engage in indirect forms of bullying (e.g., ignoring, rejecting, or relational aggression) victimization."
This is interesting though you should add a citation for this, otherwise if it is just another hypothesis, just say:
"Further research may test if females as a group (more than males) are more likely to engage in such indirect forms of bullying victimization (e.g., ignoring, rejecting, or relational aggression), as we hypothesize female peer phubbing may be a fresh manifestation of an older common form of bullying victimization perhaps seen more in females than males."
Response: Thank you for highlighting this issue. According to this comment, we have added citations for the statement that "Females are also more likely to engage in indirect forms of bullying (e.g., ignoring, rejecting, or relational aggression) victimization " in the manuscript.
4.
line 439:
I feel you still overstate the findings in the conclusions
you wrote: "in that these two effects were more potent in girls than in boys."
suggestion: "in that these two effects were slightly more potent in girls and almost nonexistent in boys." Correct? If not, explain in a bit more detail.
Plus, I would like to see a sentence somewhere (either in the conclusions or in the earlier caveat section) saying that given these weak categorical associations (of male and female), that other factors on the macro-level of the school organizational environment or the micro-level of individual psychology may be a better route to helping people, with either school based changes or with individually-tailored psychological consulting and individually tailored interventions, or both, instead of presuming you have to have only centralized managerial and categorical/sterotypical based interventions where you divide people in to male and female to give then the 'only two' (sic) treatments.
Response: We have revised the findings in the conclusions to highlight that the direct and indirect effects of peer phubbing on mobile social media addiction were only relatively more potent in girls than in boys (the direct effect was almost nonexistent in boys). We also added a sentence in the “limitation” section saying that other factors on the macro-level of the school environment or the micro-level of individual psychology may be a better route to helping people deal with peer phubbing and mobile social media addiction. Future research may explore other protective factors with strong buffering effects to generate more practical interventions.
5.
line 439:
from this: "The findings highlight how peer phubbing is directly and indirectly associated with adolescent mobile social media addiction and for whom the direct and indirect effects of peer phubbing are potent."
to this suggestion:
"The findings highlight how peer phubbing is directly and indirectly associated with more female adolescent mobile social media addiction, and how female direct and indirect effects of peer phubbing are more potent on females than on boys, when comparing averages of both gender groups."
Correct? If not, explain a bit more, and you can of course adjust the suggested wording.
Response: Thank you very much for this suggestion. We have revised the mentioned sentence above as follows: The findings highlight how peer phubbing is directly and indirectly associated with adolescent mobile social media addiction in general and how the direct and indirect effects of peer phubbing are more potent in girls than in boys, when comparing averages of both gender groups.
6.
suggested title adjustment
from this:
"Peer Phubbing and Adolescent Mobile Social Media Addiction: Testing the Roles [what roles?] of Loneliness and Gender"
to the suggestion:
Do Loneliness and Gender Act as Moderators Between Peer Phubbing and Adolescent Mobile Social Media Addiction?
What do you think? I just think the old title could be more succinct and clear of what is being tested since in the suggested version, the discussion of the IV as a mediator comes first, then using a verb on the DV, instead of a colon splitting the title into two indirect parts and leaving the 'role' between the two phrases quite undefined. Adjust using your own ideas or use this idea, yet please, adjust it somehow, I suggest.
Response: Thank you very much for your suggestion on the title. We agree with your opinion that the title should be more succinct and clear of what is being tested. According to this comment, we have changed the original title into “The mediating role of loneliness and the moderating role of gender between peer phubbing and adolescent mobile social media addiction.” The new title can clearly highlight the specific roles of loneliness and gender in the relationship between peer phubbing and adolescent mobile social media addiction.